materials science/nanotechnology

BNT-BT, hydrothermal synthesis, sintering, perovskite, dielectric properties

**Author for correspondence:**
Andrea Nesterović
e-mail: nesterovic.andrea@gmail.com

# Structure and dielectric properties of (1-x)Bi$_{0.5}$Na$_{0.5}$TiO$_3$$_{-x}$BaTiO$_3$ piezoceramics prepared using hydrothermally synthesized powders

Andrea Nesterović[1], Jelena Vukmirović[1], Ivan Stijepović[1], Marija Milanović[1], Branimir Bajac[2], Elvira Tóth[3], Željka Cvejić[3] and Vladimir V. Srdić[1]

[1]Faculty of Technology Novi Sad, University of Novi Sad, Bulevar Cara Lazara 1, Novi Sad, Serbia
[2]Institute Biosense, University of Novi Sad, Dr Zorana Đinđića 1, Novi Sad, Serbia
[3]Faculty of Sciences, University of Novi Sad, Trg Dositeja Obradovića 3, Novi Sad, Serbia

AN, 0000-0002-7001-1499; IS, 0000-0001-6439-5804

The influence of different processing parameters and various Ba$^{2+}$ addition (up to 10 mol%) on the structure and dielectric properties of Bi$_{0.5}$Na$_{0.5}$TiO$_3$-BaTiO$_3$ (BNT-BT) ceramics was investigated. The powders were hydrothermally synthesized in the alkaline environment at 180°C for different time periods. X-ray diffraction confirmed the presence of dominant rhombohedral Bi$_{0.5}$Na$_{0.5}$TiO$_3$ phase and a small amount of secondary pyrochlore Bi$_2$Ti$_2$O$_7$ phase in the pure BNT powders. In addition, one-dimensional Na$_2$Ti$_2$O$_7$ structure was also observed in the powder hydrothermally treated for a long time (i.e. 48 h). The amount of secondary pyrochlore phase in the BNT-BT powders increases with the increase of Ba$^{2+}$ content. The synthesized powders were pressed into pellets and finally sintered at various temperatures up to 1150°C. High density (more than 90%TD) was obtained in all BNT-BT sintered samples. Optimal sintering parameters were chosen in order to obtain dense ceramics with the optimal phase composition. The temperature dependence of dielectric properties for the BNT-BT ceramics was also studied. Relaxor behaviour of BNT-based ceramics and broad transition peaks are evident in all samples. Dielectric constant up to 400 as well as an acceptable low dielectric loss at temperatures lower than 200°C were obtained in BNT-BT ceramics.

This article has been edited by the Royal Society of Chemistry, including the commissioning, peer review process and editorial aspects up to the point of acceptance.

# 1. Introduction

Piezoelectric ceramics have been used for a long time, especially in the field of electronics for devices such as sensors, actuators and transducers [1]. Driven by the need for using alternative, environmentally friendly and low-cost energy sources, piezoceramics have an important role in energy harvesting. Piezoelectric energy harvesting implies collecting mechanical energy, usually from vibrations, and converting it into electrical energy, which can be stored and used for powering other devices [2]. Lead-based materials such as lead zirconate titanate ($Pb(Zr_{1-x}Ti_x)O_3$ or PZT) have been dominant piezoceramics for a long period of time, due to their superior piezoelectric and dielectric properties, easy preparation and low cost [1,3]. However, due to the toxicity of lead, there is a high tendency to replace them with environmentally friendly alternative materials [4]. Among lead-free materials, bismuth sodium titanate ($Bi_{0.5}Na_{0.5}TiO_3$ or BNT) has been recognized as a prospective replacement for PZT. BNT is a relaxor ferroelectric discovered by Smolenskii *et al.* [5] in 1960. It has an A-site substituted perovskite structure and diffuse phase transition from rhombohedral (−268–255°C) to tetragonal phase (400–500°C) and from tetragonal to cubic phase (greater than 540°C). The coexistence of rhombohedral and tetragonal phases was confirmed in the range of 255–400°C and tetragonal and cubic in the range of 500–540°C [6]. BNT shows the maximum of permittivity at about $T_m = 320$°C [1]. The structure of BNT is still intensively discussed and investigated due to its complexity. BNT ceramics exhibits piezoelectric coefficient $d_{33} = 57$–$70$ pC/N, remnant polarization $P_r = 38$ μC cm$^{-2}$ and coercive field $E_c = 73$ kV cm$^{-1}$ [7,8]. Due to the inferior functional properties in comparison with PZT, pure BNT is often modified with other perovskite materials. Takenaka *et al.* [9] reported that $(1\text{-}x)Bi_{0.5}Na_{0.5}TiO_3$–$x BaTiO_3$ (BNT-BT) solid solution shows significantly improved piezoelectric and dielectric properties near the morphotropic phase boundary (MPB) at $x = 0.06$.

BNT-based ceramics obtained by the conventional solid-state method have been intensively investigated over the years. However, due to high-temperature calcination and sintering, repeated grinding and volatilization of bismuth and sodium, other synthesis procedures such as sol–gel, co-precipitation and hydrothermal synthesis were used. Hydrothermal synthesis provides a lot of advantages in comparison with the solid-state method, such as low-temperature reaction, easier control of particle size and preparation of different morphologies such as nanoparticles and nanofibres, better dissolving of salts, and it is also a low-cost technique [10,11]. Hydrothermally synthesized bismuth sodium titanates in the alkaline environment were first obtained by Lencka *et al.* [12] with equilibrium calculations in Na–Bi–Ti–$H_2O$ system. The influence of different parameters of hydrothermal synthesis (temperature, time, NaOH concentration) on the morphology of BNT powders was investigated by different authors [10,13–16]. However, only a few published articles investigated the sintering of BNT and BNT-BT powders obtained by the hydrothermal method [17–21] and the influence of sintering parameters on microstructure and functional properties of obtained ceramics is still not completely defined. In addition, different dielectric properties of the sintered ceramics were reported. Thus, Mahmood *et al.* [19] measured dielectric properties of BNT, BNT-6BT and BNT-7BT ceramics and obtained relatively low dielectric constant values (up to 100) in the temperature range up to 500°C. Reshetnikova *et al.* [21] reported that the dielectric constant for the pure BNT is 360 due to the presence of secondary phases. Wang *et al.* [22] reported a higher value of dielectric permittivity (479) of BNT, but they also used higher synthesis time (48 h) and slightly higher sintering temperature (1130°C).

The focus of this work is the fabrication of dense BNT-BT ceramics (from hydrothermal powders) with desirable phase composition and good functional properties. Unlike the solid-state method, the main challenge with the hydrothermal method is to obtain the desired phase, since the presence of different secondary phases in samples has been noticed [23–25]. Having that in mind, the first part of the paper shows the influence of different hydrothermal synthesis conditions on the phase composition and morphology of BNT powders. The second part of the paper shows how different sintering parameters (such as time and temperature) and the addition of $Ba^{2+}$ ions have an influence on the structure and dielectric properties of BNT-based ceramics. Hence, we combined investigations of processing parameters and structural and functional properties of hydrothermally obtained BNT and BNT-BT powders in order to further explore their interdependence.

# 2. Material and methods

## 2.1. Processing of BNT-based ceramics

Pure bismuth sodium titanate ($Bi_{0.5}Na_{0.5}TiO_3$) powders were prepared by the hydrothermal method. Bismuth nitrate pentahydrate ($Bi(NO_3)_3 \cdot 5H_2O$), sodium hydroxide (NaOH) and tetrabutyl titanate

**Table 1.** Sample labelling.

| chemical formula | sample name |
| --- | --- |
| $Bi_{0.5}Na_{0.5}TiO_3$ | BNT |
| $0.98Bi_{0.5}Na_{0.5}TiO_3–0.02BaTiO_3$ | BNT-2BT |
| $0.96Bi_{0.5}Na_{0.5}TiO_3–0.04BaTiO_3$ | BNT-4BT |
| $0.94Bi_{0.5}Na_{0.5}TiO_3–0.06BaTiO_3$ | BNT-6BT |
| $0.92Bi_{0.5}Na_{0.5}TiO_3–0.08BaTiO_3$ | BNT-8BT |
| $0.90Bi_{0.5}Na_{0.5}TiO_3–0.10BaTiO_3$ | BNT-10BT |

($C_{16}H_{36}O_4Ti$) were used as starting materials. In the first step, 2.91 g of $Bi(NO_3)_3·5H_2O$ was dissolved in 30 ml of distilled water with a small addition of $HNO_3$ in order to obtain a clear solution. Second, a stoichiometric amount of tetrabutyl titanate was added slowly into the previously prepared solution under constant stirring. The molar ratio of $Bi(NO_3)_3$/ $C_{16}H_{36}O_4Ti$ was 0.5 : 1. Then, 60 ml of 10 M NaOH was introduced into the mixture drop by drop and stirred for 30 min. The mixture was transferred to a stainless steel autoclave with a capacity volume of 120 ml. Hydrothermal synthesis was carried out at 180°C for different periods of time: 6, 20 and 48 h. The mixture was cooled down to room temperature naturally, washed with distilled water until pH = 7 and finally dried at 100°C for 20 h.

Bismuth sodium titanate modified with barium titanate (($1-x)Bi_{0.5}Na_{0.5}TiO_3–xBaTiO_3$, where $x = 0$, 0.02, 0.04, 0.06, 0.08 and 0.1) powders were synthesized using the same procedure in which barium nitrate ($Ba(NO_3)_2$) was used as a source of $Ba^{2+}$ ions and duration of hydrothermal treatment was 6 h. After synthesis, the powders were washed until neutral pH was reached and dried overnight at 70°C and 4 h at 120°C. The obtained powders were uniaxially pressed into pellets under the pressure of 320 MPa. The resulting pellets were finally sintered at various temperatures up to 1150°C in the air with a heating rate of 5°C/ min. Sample labelling is shown in table 1.

## 2.2. Characterization

Phase analysis of the obtained samples was performed by X-ray diffraction (Rigaku MiniFlex 600 diffractometer) using Cu-K$\alpha$ radiation in the range from 10 to 70° with a step of 0.03° and dwell time of 3 s. Raman spectra were recorded using a Thermo Scientific DXR Raman Microscope with 780 nm wavelength laser. The morphology and microstructure of the synthesized and sintered samples were observed by scanning electron microscopy ((JEOL, JSM 6460LV). An LCR device (LCR-8101 1 MHz Precision LCR Meter, GW Instek) was used for dielectric measurements of the sintered ceramic samples at temperatures up to 370°C in the frequency range from 50 kHz up to 1 MHz.

# 3. Results and discussion

## 3.1. Phase analysis and microstructure of BNT-based powders

The influence of time of hydrothermal synthesis on the phase composition and morphology of the pure $Bi_{0.5}Na_{0.5}TiO_3$ (BNT) was investigated. XRD patterns of the $Bi_{0.5}Na_{0.5}TiO_3$ powders synthesized at 180°C for different times (6, 20 and 48 h) are presented in figure 1. All diffraction peaks are assigned to the rhombohedral $Bi_{0.5}Na_{0.5}TiO_3$ phase according to JCPDS card No. 36-0340. The observed broad and asymmetric peaks were formed due to the overlapping of the dominant $Bi_{0.5}Na_{0.5}TiO_3$ phase with XRD peak of the secondary phase. Thus, the presence of $Bi_2Ti_2O_7$ with pyrochlore-type structure (JCPDS card No. 32-0118) is confirmed. In addition, the XRD pattern of the sample synthesized for 48 h contains a new peak at 10.6° assigned to the $Na_2Ti_2O_7$ phase (according to JCPDS card No. 31-1329) and can be related to the presence of one-dimensional structures. High pressure and longer time of hydrothermal synthesis leads to the breaking of weak Ti–O bonds which are attacked by $Na^+$ and $OH^-$, followed by the formation of the nanosheets. These nanosheets have a tendency to curl at their edges which leads to the formation of one-dimensional nanotubes. This mechanism was already recognized and explained in our previous papers [26,27].

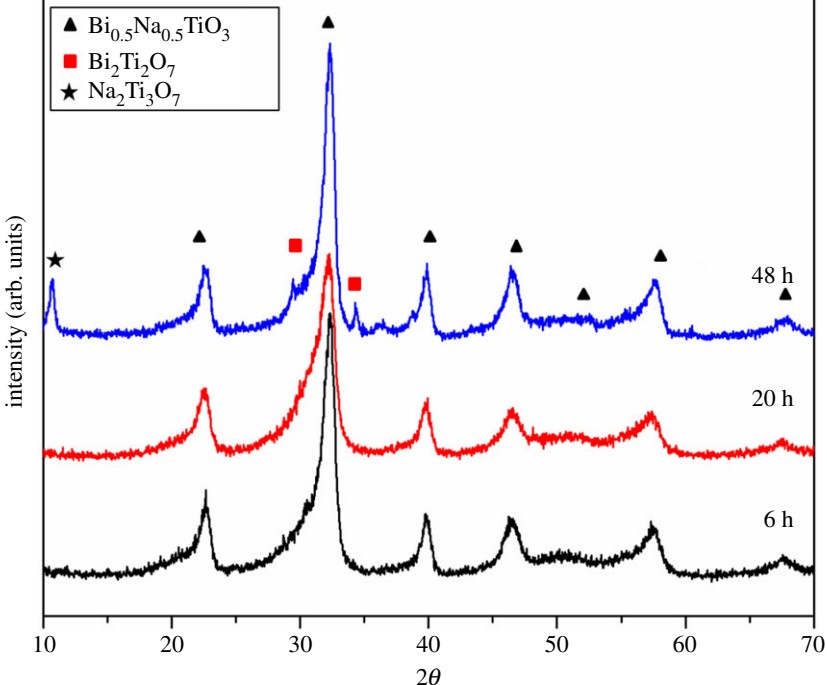

**Figure 1.** XRD patterns of $Bi_{0.5}Na_{0.5}TiO_3$ powders hydrothermally prepared using different synthesis times.

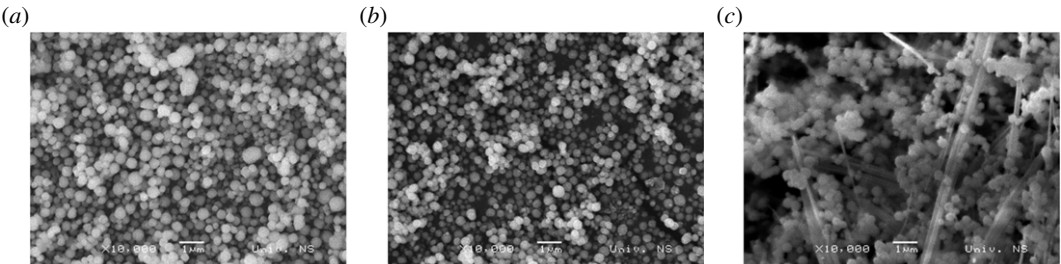

**Figure 2.** SEM images of $Bi_{0.5}Na_{0.5}TiO_3$ powders synthesized for (*a*) 6 h, (*b*) 20 h and (*c*) 48 h.

Spherical nanoparticles with the average size of the particles in the range 100–350 nm are observed (figure 2) in the case of the samples synthesized for 6 and 20 h without a noticeable difference between them. On the other hand, by increasing the synthesis time to 48 h, changes in morphology can be noticed. There is an evidence of both spherical nanoparticles and nanofibres in the sample, in accordance with the XRD results. Based on SEM images, it can be concluded that the longer time of hydrothermal synthesis leads to the formation of nanofibres.

## 3.2. Characterization of BNT-BT-based ceramics

### 3.2.1. Characterization of pure BNT

Since the similar and desired phase composition and morphology were obtained for the samples synthesized for 6 and 20 h, the powder synthesized for 6 h was used for future investigation of its behaviour at high temperatures. Thus, XRD patterns of the pure BNT sintered at 1050°C, 1100°C and 1150°C for 1 h are shown in figure 3. In all samples, the perovskite BNT phase (JCPDS card No. 36-0340) is present as the primary phase. The presence of the secondary pyrochlore $Bi_2Ti_2O_7$ phase (JCPDS card No. 32-0118) has also been noticed in all samples. The XRD pattern of the sample sintered at 1150°C contains additional peaks which are assumed to be non-stoichiometric titanate phase, confirming that temperature of 1100°C is optimal for the thermal treatment.

The samples were also investigated by Raman spectroscopy (figure 4). When the temperature of sintering is increased, the intensity of Raman peaks is decreased, due to the higher amount of

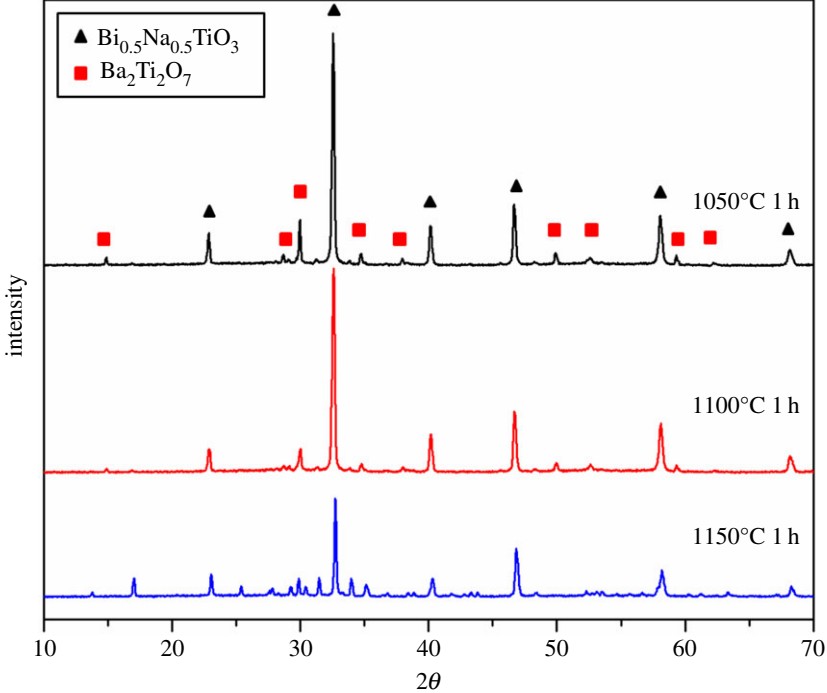

**Figure 3.** XRD patterns of BNT ceramics sintered at different temperatures.

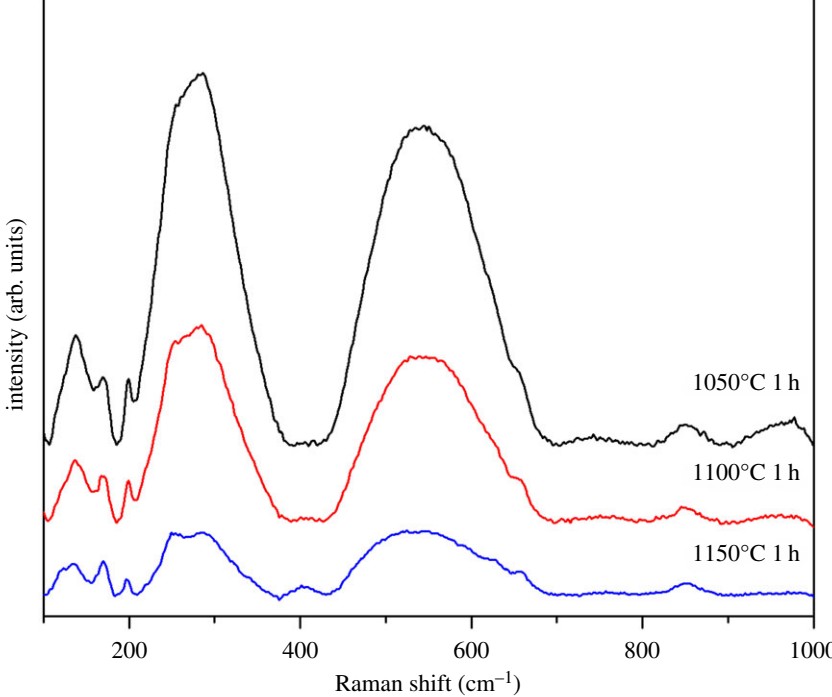

**Figure 4.** Raman spectra of BNT ceramics sintered at different temperatures.

secondary phases. Broad peaks are a consequence of the overlapping of peaks and disorder of A-site caused by the presence of different $Bi^{3+}$ and $Na^+$ ions [28]. Peaks in the range of 100 to 200 $cm^{-1}$ are assigned to the vibration of Na–O and Bi–O bonds. The peak at 137 $cm^{-1}$ is associated with Na–O bond vibrations. The intensive peak at 286 $cm^{-1}$ is present due to vibrations of $TiO_6$ octahedra. It can be noticed that this peak splits into two different peaks when the sintering temperature increases, which can be due to different vibrations of Ti–O bonds in $Bi_{0.5}Na_{0.5}TiO_3$ and $Bi_2Ti_2O_7$ phases [29]. Peaks present in the range of 413 to 820 $cm^{-1}$ are dominated by the oxygen atom vibrations [30,31]. The peak at 851 $cm^{-1}$ can be correlated to the presence of oxygen vacancies [32].

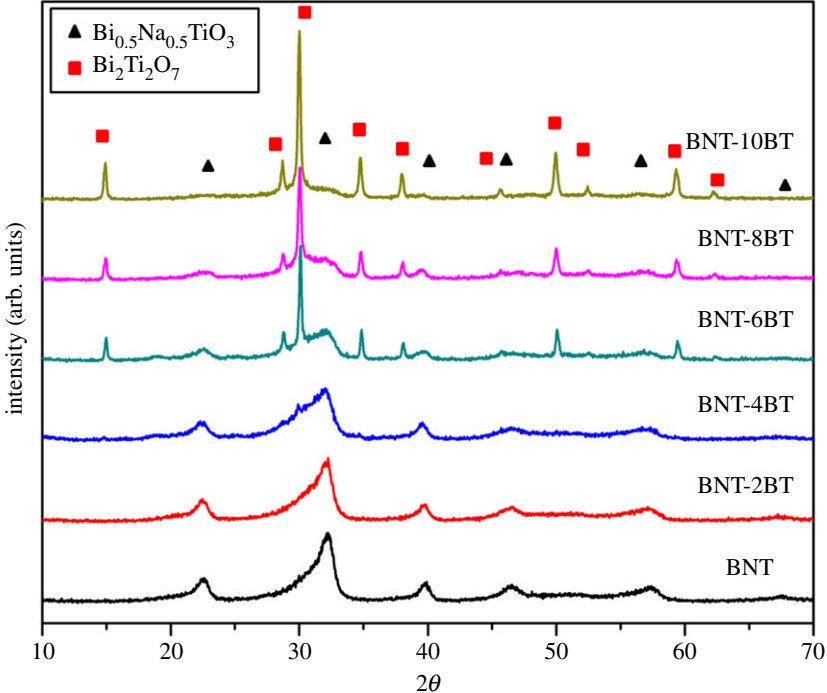

**Figure 5.** XRD patterns of BNT-BT powders after synthesis.

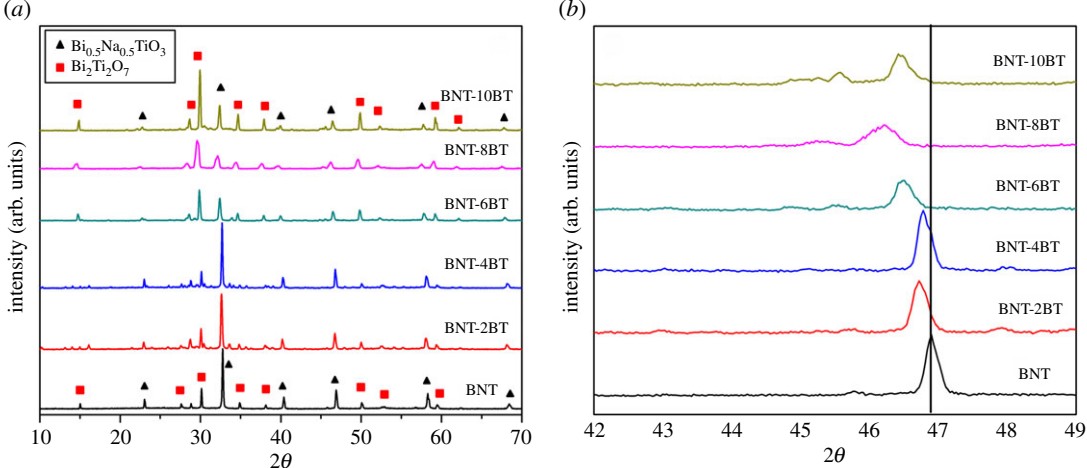

**Figure 6.** XRD of BNT-BT ceramics after sintering at 1100°C/1 h for 2θ in the range of (*a*) 10–70° and (*b*) 42–49°.

### 3.2.2. Characterization of BNT-BT ceramics

After the optimization of processing parameters for the pure BNT, the influence of addition Ba$^{2+}$ ions on the structure, density and dielectric properties of the (1-$x$)Bi$_{0.5}$Na$_{0.5}$TiO$_3$–$x$BaTiO$_3$ sintered ceramics was investigated. Figure 5 shows XRD of the as-synthesized BNT-BT powders. It can be seen that the perovskite BNT phase is dominant in the samples BNT, BNT-2BT and BNT-4BT while in the sample with higher BT content, it is present as a secondary phase. Thus, it can be concluded that increased BT content stabilizes the pyrochlore phase.

Similar results were obtained after sintering at 1100°C/1 h (figure 6*a*), i.e. the perovskite BNT phase is present as a primary phase for low content of BT, while the pyrochlore phase becomes dominant when BT content is greater than or equal to 6 mol%. The single peak between 45° and 48° characterizes the rhombohedral structure of BNT, and it is present in the samples BNT, BNT-2BT and BNT-4BT (figure 6*b*). When the amount of BT is 6 mol% and higher, additional peaks could be noticed, which can be assigned to the presence of the tetragonal phase. Also, these peaks are shifted to the lower

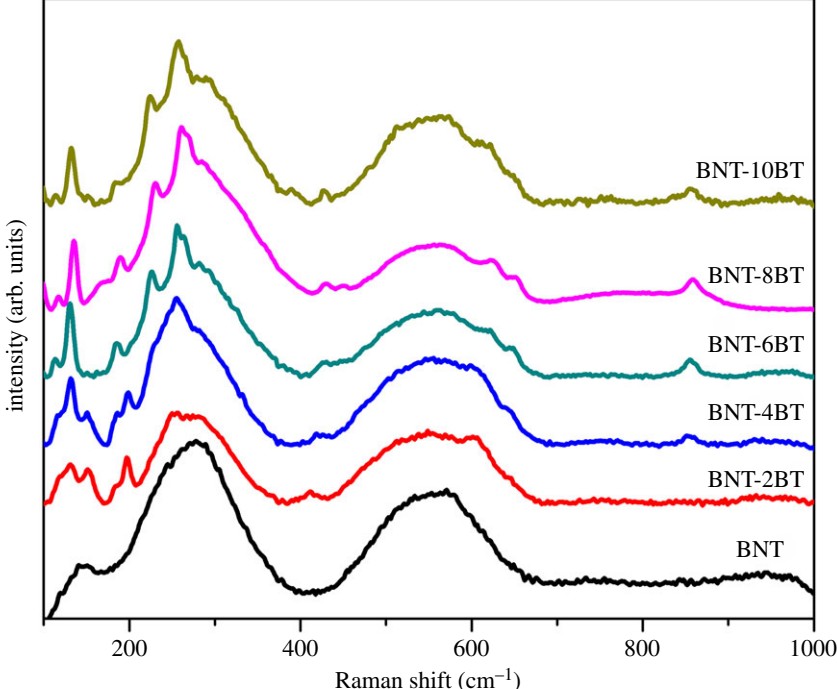

**Figure 7.** Raman spectra of BNT-BT ceramics after sintering at 1100°C/1 h.

angles as the amount of BT increases, due to the different lattice parameters of BT and BNT. It can be seen that besides the dominant pyrochlore phase, the coexistence of rhombohedral and tetragonal phases is observed for the samples BNT-6BT, BNT-8BT and BNT-10BT, which is previously confirmed in several publications [33,34].

Raman spectra of the BNT-BT ceramics after sintering at 1100 °C/1 h are shown in figure 7. In comparison with the Raman spectra of the pure BNT, it can be seen that the peak at 280 cm$^{-1}$ splits into three different peaks (290, 260 and 231 cm$^{-1}$) which is caused by different Ti–O vibrations of $Bi_{0.5}Na_{0.5}TiO_3$ and $Bi_2Ti_2O_7$ phases (see §4.2.1) and addition of $Ba^{2+}$ ions. The splitting of these peaks becomes more intensive with the increase in the amount of BT content. Also, the additional peaks in the range from 630 to 655 cm$^{-1}$ are associated with the presence of $Ba^{2+}$ ions in the structure and the increase of the average ionic radius size in the lattice of the BNT-BT ceramics. The peak at 860 cm$^{-1}$ indicated that the oxygen vacancies are present in the BNT-BT samples with 4–10% BT. [29,32]

SEM micrographs of the fresh fracture of the BNT-BT ceramics after sintering at 1100°C/1 h are shown in figure 8. The image inserted in figure 8a shows the surface of the pure BNT ceramics with well-defined grains and the average size of approximately 1.5 μm. It is evident that high density (more than 90% TD) is obtained in all sintered samples. However, a small amount of pores are present with sizes below 1 μm. Also, the presence of colour nuances could be related to the presence of two different phases, especially for the sintered samples with a higher amount of secondary phase, which is confirmed by the XRD.

## 3.3. Dielectric properties of BNT-BT ceramics

Temperature dependence of dielectric properties for the BNT, BNT-2BT, BNT-4BT, BNT-6BT, BNT-8BT and BNT-10BT ceramics was investigated up to 375°C and in the frequency range from 50 kHz to 1 MHz (figures 9–11). Below 200°C dielectric constant for all samples is up to 400, which is relatively low in comparison to the ceramics obtained from solid-state synthesized powder. The possible explanation is the existence of the pyrochlore phase, smaller grain size and more grain boundaries. The dielectric constant versus temperature curves for all measured samples exhibit broad peaks, which indicates a diffuse phase transition characteristic for relaxor ferroelectric materials. It is shown that the dielectric constant for the pure BNT increases until its broad maximum value at around 320°C, which corresponds to the transition of antiferroelectric to paraelectric phase or temperature of maximum dielectric constant $T_m$ [35]. Another anomaly is noticed in the BNT-BT ceramics in the range from 150°C to 200°C, which is defined as ferroelectric to antiferroelectric phase transition or depolarization temperature $T_d$ [29].

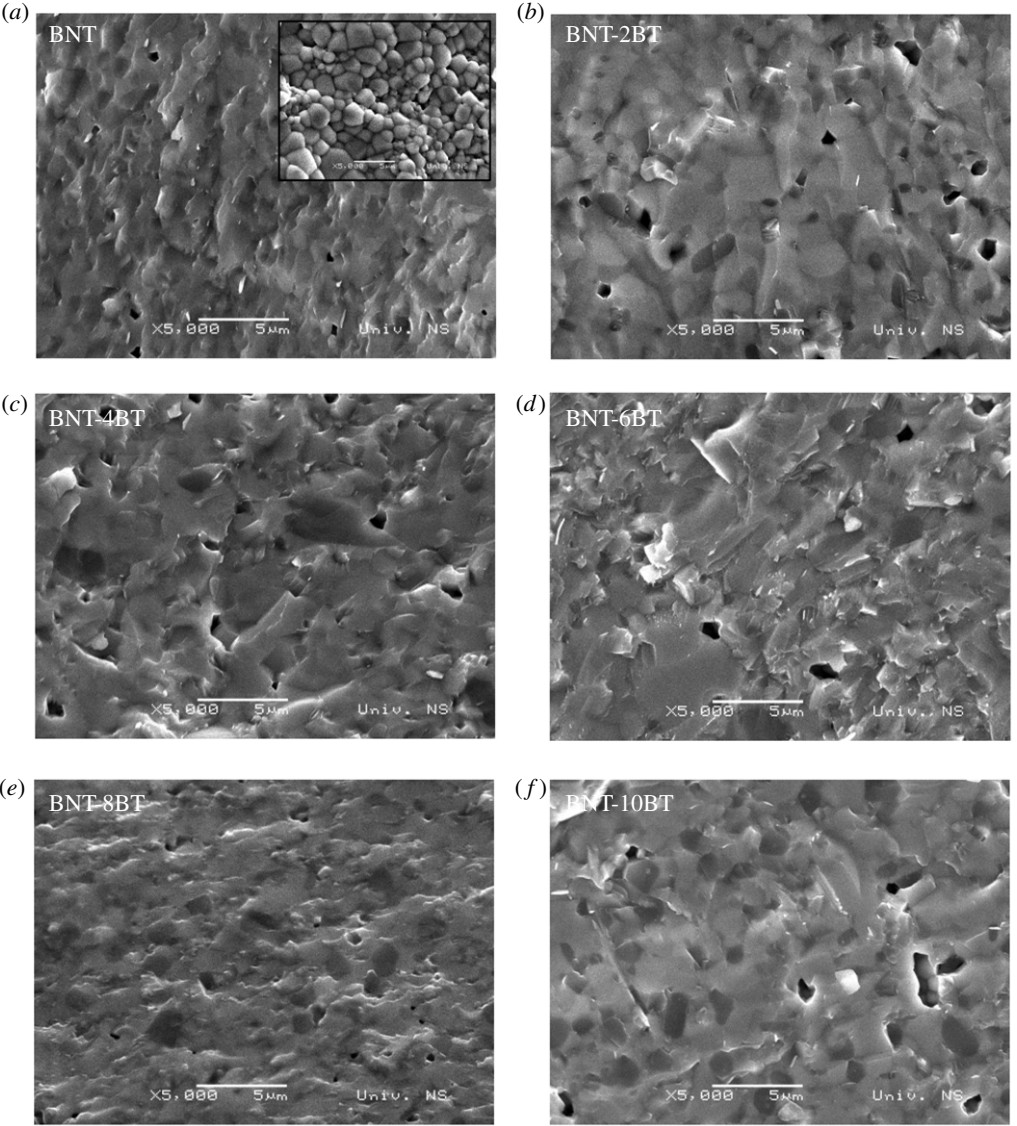

**Figure 8.** SEM images of (*a*) BNT, (*b*) BNT-2BT, (*c*) BNT-4BT, (*d*) BNT-6BT, (*e*) BNT-8BT and (*f*) BNT-10BT ceramics.

Figure 10 shows the temperature dependence of the dielectric constant for all BNT-BT samples. The maximum dielectric constant for the pure BNT is 398 at the transition temperature ($T_m$), and it increases for the BNT-2BT and BNT-4BT. Furthermore, the trend in decrease of dielectric constant is noticed in the samples when BT content is greater than or equal to 6 mol%. These results are in accordance with XRD results shown in figure 6, which show that the dominant phase in the pure BNT, BNT-2BT BNT-4BT is the perovskite BNT phase. With the further increase of BT, the pyrochlore $Bi_2Ti_2O_7$ phase becomes dominant. The pyrochlore phase has a cubic crystal structure which impairs functional properties of materials. Also, by the addition of BT up to 6 mol%, it is noticed that the maximum value of the dielectric constant ($T_m$) is shifted to the lower values of the temperature. The BNT-8BT and BNT-10BT show diffuse transitions at higher temperatures (between 320 and 350°C) which is also connected with the phase composition. Below 200°C dielectric losses (figure 11) are below 0.1 for all BNT-BT ceramics in the measured frequency range. It is shown that dielectric loss values increase at higher temperatures which can be associated with thermally activated conductivity [36].

## 4. Conclusion

In this work, bismuth sodium titanate-based powders were successfully synthesized by the hydrothermal method at 180°C. It is shown that different time of synthesis has an influence on the phase composition and morphology of $Bi_{0.5}Na_{0.5}TiO_3$ powders. The powder synthetized for 6 h showed the presence of desired crystal BNT phase and spherical nanoparticles with a narrow size

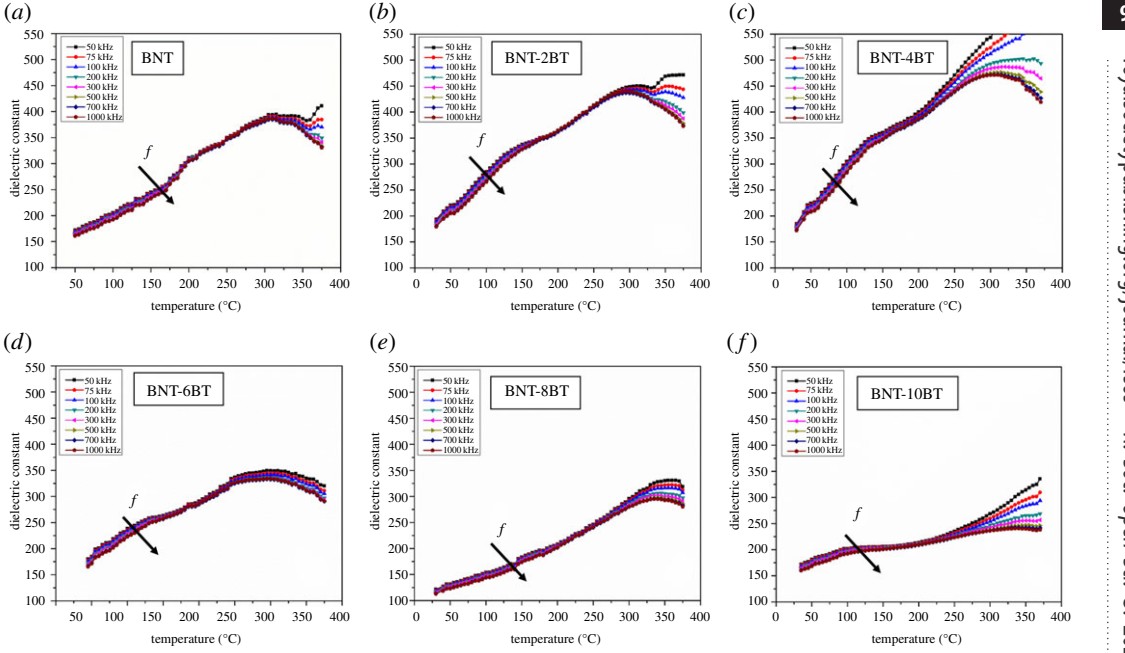

**Figure 9.** Temperature dependence of dielectric constant for (*a*) sintered BNT, (*b*) BNT-2BT, (*c*) BNT-4BT, (*d*) BNT-6BT, (*e*) BNT-8BT and (*f*) BNT-10BT ceramics.

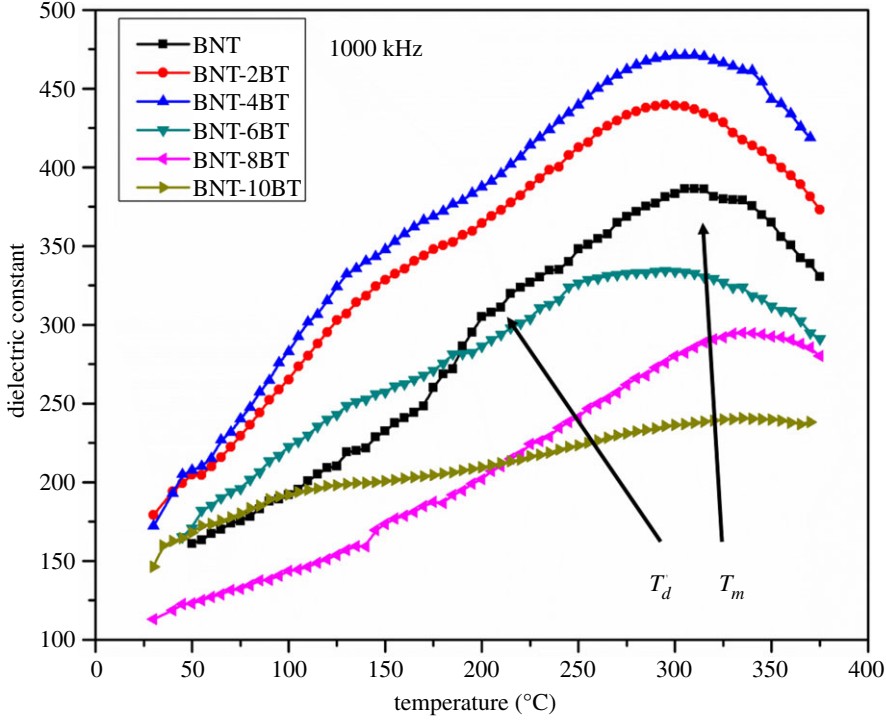

**Figure 10.** Temperature dependence of dielectric constant for pure BNT and BNT-BT samples at 1000 kHz.

distribution. Furthermore, the powder was pressed and sintered at different temperatures up to 1150°C in order to follow the phase changes and sinterability of $Bi_{0.5}Na_{0.5}TiO_3$ ceramics. The temperature of 1100°C was chosen as the optimal temperature for thermal treatment of all samples due to the presence of dominant perovskite BNT phase and minimized secondary phases which are confirmed by XRD. The results obtained by Raman spectroscopy are in correlation with XRD and the Bi–O, Na–O and $TiO_6$ bond vibrations are present in the sintered ceramic sample.

XRD analysis of the BNT-BT powders and sintered ceramics showed that the rhombohedral BNT phase is the dominant phase for the samples with the low content of BT, and for 6 mol% of BT and

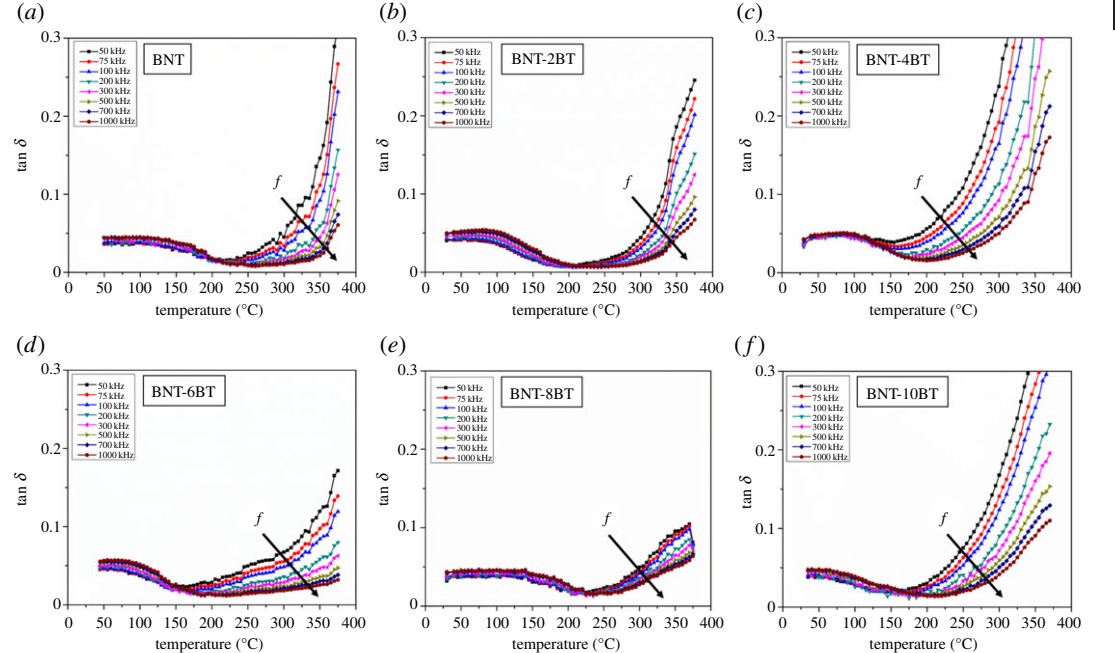

**Figure 11.** Temperature dependence of dielectric loss for (*a*) sintered BNT, (*b*) BNT-2BT, (*c*) BNT-4BT, (*d*) BNT-6BT, (*e*) BNT-8BT and (*f*) BNT-10BT ceramics.

higher, the pyrochlore phase becomes the primary phase. Also, the shift of the single peak between 45° and 48° confirms that $Ba^{2+}$ ions entered the structure. The SEM micrographs confirmed that high density (more than 90% TD) is obtained in all sintered BNT-BT samples. The dielectric measurements of the BNT-BT sintered ceramics showed that the pure BNT has a broad maximum of dielectric constant at around 320°C typical for relaxor ferroelectrics. The dielectric constant increases and the phase transition is shifted to the lower temperatures for the BNT-2BT and BNT-4BT. The dielectric constant decreases for the samples with higher content of barium (BNT-6BT, BNT-8BT and BNT-10BT) where the dominant pyrochlore phase is confirmed by XRD.

Ethics. Research on humans must include a statement detailing ethical approval and informed consent. Research using animals must adhere to local guidelines and state that appropriate ethical approval and licences were obtained. Please read our editorial policies carefully before submission.

Data accessibility. Datasets for all figures are available from the Dryad Digital Repository: https://doi.org/10.5061/dryad. kh189324r [37].

Authors' contributions. A.N. carried out the laboratory work, measured and analysed dielectric measurements and drafted the manuscript. J.V. participated in the laboratory work, analysed the dielectric measurement and helped in manuscript drafting. B.B. helped with the analysis of dielectric measurements and SEM images. E.T. and Ž.C. measured and helped with the analysis of Raman spectroscopy. I.S. measured and analysed XRD spectra and critically revised the manuscript. M.M. and V.V.S. coordinated the study and critically revised the manuscript. All authors gave final approval for publication and agree to be held accountable for the work performed therein.

Competing interests. We declare we have no competing interests

Funding. This work was financially supported by the Ministry of Education, Science and Technological Development of the Republic of Serbia, Project No. III45021 and Project program 451-03-68/2020-14/200053.

Acknowledgements. The authors would like to thank Miloš Bokorov from Department of Biology and Ecology, Faculty of Sciences, University of Novi Sad for professional help with the SEM measurements.

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
