## [Peer Review File · Royal Society Open Science]

Review History

RSOS-202365.R0 (Original submission)

Review form: Reviewer 1

Is the manuscript scientifically sound in its present form?

No

Are the interpretations and conclusions justified by the results?

No

Is the language acceptable?

Yes

Do you have any ethical concerns with this paper?

No

Have you any concerns about statistical analyses in this paper?

No

Recommendation?

Major revision is needed (please make suggestions in comments)

Comments to the Author(s)

Review comments_RSOS-202365

The authors have reported entitled on Structure and dielectric properties of $(1-x)$ Bi_{0.5}Na_{0.5}TiO₃- x BaTiO₃ (BNT-BT) piezoceramics prepared using hydrothermally synthesized powders. The manuscript has solely characterization techniques such as XRD, FESEM, and Dielectric studies are just routine, and there is either no novelty or application in this work, it can be accepted only after addressing the following comments.

1. The authors reported the influence of different processing parameters and various BT compositions. It is very difficult to understand with the investigation of many processing parameters (different synthesis times, different sintering temperatures and different BT compositions), which leads to much confusion for readers.
2. I suggest you to report the influence of BNT-BT compositions (only one parameter) on structural, microstructural and dielectric properties in detail along with optimization of sample preparation.
3. The Raman spectra of BNT-BT studies are more important than the sintered samples.
4. The XRD studies of BNT-BT ceramics sintered at 1100 oC for 1h also show pure phase along with secondary phase even after optimization of many processing parameters.
5. The appearance of secondary phase is due to the volatilization of Bi and Na elements, which can be controlled with the excess amount of Bi and Na into BNT. There are many reports in BNT-BT based ceramics with pure phase. So, what is the novelty in this work?
6. All samples FESEM micrographs should be like Insert image of Fig. 7(a): follow same scale and magnification.
7. The XRD and FESEM investigated for BNT to BNT- 10BT, but dielectric properties investigated only for BT, BNT-2BT and BNT-6BT samples. It should be same for all the samples to understand investigated properties systematically.
8. In Fig. 8(a), It seems to be there is no transition temperature even at high frequencies. Normally, the BNT has two phase transition temperatures: TC at 320 oC and Td at 200 oC.
9. In Fig. 8(a), all curves look overlapped. It looks better if authors represent in line instead of symbol and line.
10. All the samples, the transition peak is very broad. How did you define transition temperature? Is there any method to estimate the TC ?
11. The data of dielectric constant of BNT-2BT missing below 100 0C and above 330 0C.
12. The addressing of dielectric properties (in terms of dielectric constant and loss values) with the substitution of BT is mandatory.
13. There is no comparison of investigated properties with earlier reports.

Review form: Reviewer 2 (Cleocir Dalmaschio)

Is the manuscript scientifically sound in its present form?

Yes

Are the interpretations and conclusions justified by the results?

Yes

Is the language acceptable?

Yes

Do you have any ethical concerns with this paper?

Yes

Have you any concerns about statistical analyses in this paper?

No

Recommendation?

Accept with minor revision (please list in comments)

Comments to the Author(s)

Dear authors,

The manuscript presents interesting results and discussion.

Besides that, some point can be improved before publication, as highlighted below.

- 1 - Some keywords have same word in title. I would like suggest use in different keywords from title;
- 2 - summary / abstract is not numbered in RSOS papers;
- 3 - Section "3.1. Processing of BNT based ceramics" should receive special attention. Improve synthesis details, e.g. heating rate in hydrothermal and sintering process. strange sentences "constant stirring on magnetic stirrer." ...Insert synthesis details;
- 4 - in Page 4 of 12, line 36, was presented "relatively narrow size distribution" relative to what? statistical analysis was used to indicate narrow size distribution?
- 5 - In XRD, especially Fig. 1, asymmetric peaks are observed. Probably there is something not adequate in XRD equipment or sample preparation;
- 6 - Page 10 of 12, line 55, TD definition was not indicated. Please, revise all text and define abbreviations;
- 7 - In 34 references, only 2 were published in last 3 years. Is it a topic well explored, that goes to burial ground or authors did not follow current publications?

Decision letter (RSOS-202365.R0)

Dear Ms Nesterovic:

Title: Structure and dielectric properties of $(1-x)\text{Bi}_0.5\text{Na}_0.5\text{TiO}_3-x\text{BaTiO}_3$ piezoceramics prepared using hydrothermally synthesized powders

Manuscript ID: RSOS-202365

The editor assigned to your manuscript has now received comments from reviewers. We would like you to revise your paper in accordance with the referee and Subject Editor suggestions which can be found below (not including confidential reports to the Editor). Please note this decision does not guarantee eventual acceptance.

Please submit your revised paper before 07-May-2021. Please note that the revision deadline will expire at 00.00am on this date. If we do not hear from you within this time then it will be assumed that the paper has been withdrawn. In exceptional circumstances, extensions may be possible if agreed with the Editorial Office in advance. We do not allow multiple rounds of revision so we urge you to make every effort to fully address all of the comments at this stage. If deemed necessary by the Editors, your manuscript will be sent back to one or more of the original reviewers for assessment. If the original reviewers are not available we may invite new reviewers.

On behalf of the Subject Editor Professor Anthony Stace and the Associate Editor Dr Dattatray Late.

RSC Associate Editor:
Comments to the Author:
Major Revision

RSC Subject Editor:
Comments to the Author:
(There are no comments.)

Reviewers' Comments to Author:
Reviewer: 1
Comments to the Author(s)
Review comments_RSOS-202365

The authors have reported entitled on Structure and dielectric properties of $(1-x)\text{Bi}_{0.5}\text{Na}_{0.5}\text{TiO}_3-x\text{BaTiO}_3$ (BNT-BT) piezoceramics prepared using hydrothermally synthesized powders. The manuscript has solely characterization techniques such as XRD, FESEM, and Dielectric studies are just routine, and there is either no novelty or application in this work, it can be accepted only after addressing the following comments.

1. The authors reported the influence of different processing parameters and various BT compositions. It is very difficult to understand with the investigation of many processing parameters (different synthesis times, different sintering temperatures and different BT compositions), which leads to much confusion for readers.
2. I suggest you to report the influence of BNT-BT compositions (only one parameter) on structural, microstructural and dielectric properties in detail along with optimization of sample preparation.
3. The Raman spectra of BNT-BT studies are more important than the sintered samples.
4. The XRD studies of BNT-BT ceramics sintered at 1100 oC for 1h also show pure phase along with secondary phase even after optimization of many processing parameters.
5. The appearance of secondary phase is due to the volatilization of Bi and Na elements, which can be controlled with the excess amount of Bi and Na into BNT. There are many reports in BNT-BT based ceramics with pure phase. So, what is the novelty in this work?
6. All samples FESEM micrographs should be like Insert image of Fig. 7(a): follow same scale and magnification.
7. The XRD and FESEM investigated for BNT to BNT- 10BT, but dielectric properties investigated only for BT, BNT-2BT and BNT-6BT samples. It should be same for all the samples to understand investigated properties systematically.
8. In Fig. 8(a), It seems to be there is no transition temperature even at high frequencies. Normally, the BNT has two phase transition temperatures: TC at 320 oC and Td at 200 oC.
9. In Fig. 8(a), all curves look overlapped. It looks better if authors represent in line instead of symbol and line.
10. All the samples, the transition peak is very broad. How did you define transition temperature? Is there any method to estimate the TC ?
11. The data of dielectric constant of BNT-2BT missing below 100 0C and above 330 0C.
12. The addressing of dielectric properties (in terms of dielectric constant and loss values) with the substitution of BT is mandatory.
13. There is no comparison of investigated properties with earlier reports.

Reviewer: 2

Comments to the Author(s)

Dear authors,

The manuscript presents interesting results and discussion.

Besides that, some point can be improved before publication, as highlighted below.

- 1 - Some keywords have same word in title. I would like suggest use in different keywords from title;
- 2 - summary / abstract is not numbered in RSOS papers;
- 3 - Section "3.1. Processing of BNT based ceramics" should receive special attetion. Improve synthesis detaisl, e g. heating rate in hydrothemal and sinteruing process. strange sentences "constant stirring on magnetic stirrer." ...Insert synthesis details;
- 4 - in Page 4 of 12, line 36, was presented "relatively narrow size distribution" relative to what? statistical analysis was used to indicates narrow size distribution?
- 5 - In XRD, speccially Fig. 1, assimetric peaks are observed. Probably there is something not adeqaute in XRD equipament or sample preparation;

6 - Page 10 of 12, line 55, TD definition was not indicated. Please, revise all text and define abbreviattons;

7 - In 34 references, only 2 were published in last 3 years. Is it a topic well explored, that goes to burial ground or authors did not are follow current publications?

Author's Response to Decision Letter for (RSOS-202365.R0)

See Appendix A.

Decision letter (RSOS-202365.R1)

Dear Ms Nesterovic:

Title: Structure and dielectric properties of $(1-x)\text{Bi}_0.5\text{Na}_0.5\text{TiO}_3-x\text{BaTiO}_3$ piezoceramics prepared using hydrothermally synthesized powders
Manuscript ID: RSOS-202365.R1

It is a pleasure to accept your manuscript in its current form for publication in Royal Society Open Science. The chemistry content of Royal Society Open Science is published in collaboration with the Royal Society of Chemistry.

On behalf of the Subject Editor Professor Anthony Stace and the Associate Editor Dr Dattatray Late.

RSC Associate Editor
Comments to the Author:

Authors have answered all the queries raised by referee. The manuscript is suitable for publication as it is.

Reviewer(s)' Comments to Author:

Appendix A

Reviewer 1:

We highly appreciate the effort and kind suggestions that have been made by Reviewer 1 for the revision of our submitted manuscript. We have corrected the manuscript accordingly to the suggestions made in Introduction, Experimental and Results part of the manuscript.

1. and 2. In the first part of the paper it was necessary to investigate the influence of processing parameters only for the pure $\text{Bi}_{0.5}\text{Na}_{0.5}\text{TiO}_3$ (BNT). The influence of time of hydrothermal synthesis on the phase composition and morphology of the synthesized pure BNT powder, as well as the influence of processing parameters (different sintering temperature) on the sintered pure BNT ceramics were investigated. The results showed that the optimal time for the hydrothermal synthesis was 6 hours and the optimal sintering parameters were 1100 °C/1 h. After the optimization of processing parameters for the pure BNT, the second part of the paper shows how the different content of Ba^{2+} have influence on the phase composition, density and dielectric properties of $(1-x)\text{Bi}_{0.5}\text{Na}_{0.5}\text{TiO}_3-x\text{BaTiO}_3$ sintered ceramics.

3. The Raman spectra for the sintered BNT, BNT-2BT, BNT-4BT, BNT-6BT, BNT8-BT and BNT-10BT were added into manuscript as Figure 7.

4. and 5. Unlike the solid-state method, where formation of secondary phases can be minimized by adding the excess amount of Bi and Na, the hydrothermal method is more complicated because the secondary phases are present even after synthesis and before sintering. This means that the higher concentration of Ba^{2+} ions, which have larger ionic radius than Bi^{3+} and Na^+ , can disbalance the formation of BNT-BT and lead to the formation of secondary phases. Despite that, hydrothermal method is very important to investigate because it offers many advantages such as low temperature reaction, easier control of particle size and preparation of different morphologies such as nanoparticles and nanofibers, better dissolving of salts, and it is also a low-cost technique.

6. We obtained new SEM images of the surfaces but we think that it would be better off without their insertion in the manuscript. Namely, there are no new information arising for them since the surface is rough. The special attention is put on the density of the samples and the density is much better observed on the fractured surface and that is the reason why we put SEM images.

7. The authors accepted the suggestion of the reviewer and measured dielectric properties for all the samples (pure BNT, BNT-2BT, BNT-4BT, BNT-6BT, BNT-8BT, BNT-10BT). All the measurements had been done again continuously in series to avoid possible influence of conductive silver paste used for electrodes, environmental factors, measurement set-up etc. So there are some slight differences in absolute values between newly acquired results and previously shown properties in the first version of manuscript (BNT, BNT-2BT and BNT-6BT) but relative changes are small. However, the trends of the each sample are consistent with our previous results. It is also important to mention that the same ceramic samples were used for the previous and new measurements but with new electrode layer.

8. Figure 9a shows very broad peak for pure BNT, especially at the highest frequency (1 MHz). The broad peak is characteristic for relaxor-like ferroelectric materials and it is also explained in different research papers (C. Xu et al. / Solid State Sciences 10 (2008) 934e940, Current Applied Physics 12 (2012) 1100-1105, Journal of Alloys and Compounds 509 (2011) 9138– 9143, Ceramics International 42 (2016) 15664–15670, Ferroelectrics, 404:240–246, 2010]. Also, it is important to mention that ceramics obtained by wet-chemical synthesis is usually connected with formation of smaller grains, which can affect the weaker response on dielectric measurements and different shape of curve.

9. The curves in Fig. 9a are overlapped because the dielectric constant shows very close values on different frequencies. The line and symbol style is selected in order to show clearly every measure point, but we accepted the suggestion and decreased the size of the symbol. If we presented as line, these positions are not going to be visible.

10. As it is already explained in text and answer 8 above in this text, the transition peaks should be broad. The temperature of maximum dielectric constant T_m (defined also as a Curie

temperature) is defined as a temperature where the dielectric constant has a highest measured value.

11. The dielectric constant of BNT-2BT was measured again in the wider range of the temperatures.

12. The addressing of the dielectric properties with the substitution of BNT is added and explained in the results.

13. The comparison with other publications have been added in Introduction part of manuscript.

Reviewer 2:

We thank Reviewer 2 for the kind suggestions and systematically done review. We have accepted all the suggested corrections addressed by the reviewer.

1. The changes in key words are: BNT-BT and perovskite have been added and $\text{Bi}_{0.5}\text{Na}_{0.5}\text{TiO}_3-x\text{BaTiO}_3$ and structure are deleted.

2. The authors used official RSOS paper template for the manuscript preparation. If the paper is accepted for publication the editor and layout designer should change this if there is a mistake.

3. The results showed in 4.1. parts are splitted into 4.1. Phase analysis and microstructure of BNT based powders and 4.2. Characterization of BNT-BT based ceramics (4.2.1. Characterization of pure BNT and 4.2.2. Characterization of BNT-BT ceramics). The term “under constant stirring on magnetic stirrer” is changed into “into previously prepared solution under constant stirring”. The main synthesis details are inserted as well as heating rate in sintering process.

4. The term “relatively narrow size distribution” is change into “the average size of the particles is in the range 100–350 nm”

5. Asymmetric peaks observed after hydrothermal synthesis are characteristic for samples synthesized by hydrothermal synthesis and they are also reported by different authors in other research papers (Ceramics International 43 (2017) 11580–11587, Journal of Alloys and Compounds 484 (2009) 801–805, Advanced Materials Research Vol. 763 (2013) pp 135-138). The asymmetry of the peaks is still not clear.

6. The additional explanation has been added.

7. New references have been added.